# Cut-Off Points of Visceral Adipose Tissue Associated with Metabolic Syndrome in Military Men

**DOI:** 10.3390/healthcare9070886

**Published:** 2021-07-14

**Authors:** Samir Ezequiel da Rosa, Amina Chain Costa, Marcos S. R. Fortes, Runer Augusto Marson, Eduardo Borba Neves, Laercio Camilo Rodrigues, Paula Fernandez Ferreira, Jose Fernandes Filho

**Affiliations:** 1Brazilian Army Physical Fitness Research Institute (IPCFEx), Rio de Janeiro 22291-090, Brazil; profdr1.divpesq@ipcfex.eb.mil.br (M.S.R.F.); profdr2.divpesq@ipcfex.eb.mil.br (R.A.M.); eduardoneves@utfpr.edu.br (E.B.N.); paula.fernandez@eb.mil.br (P.F.F.); 2Federal Fluminense University (UFF), Niterói 24210-240, Brazil; aminacosta@id.uff.br; 3Federal University of Rio de Janeiro (UFRJ), Rio de Janeiro 21941-901, Brazil; laercio.rodrigues@eb.mil.br (L.C.R.); jff@eefd.ufrj.br (J.F.F.)

**Keywords:** visceral adipose tissue, military, metabolic syndrome, cut-off points, DXA

## Abstract

Visceral adipose tissue (VAT) has high metabolic activity and secretes a larger number of adipokines that are related to the inflammatory process. Quantifying VAT could estimate the risk of developing Metabolic Syndrome (MetS). This study was designed to determine the VAT cut-off points assessed by DXA associated with MetS in military men. In total, 270 (37.5 ± 6.9 years) military men from the Brazilian Army (BA) participated in the study. Anthropometric measurements, assessment of body composition by dual X-ray absorptiometry (DXA), hemodynamics and biochemical tests were performed. The Student’s *t* test, independent samples, Person’s correlation, ROC curve, Youden Index and positive (PPV) and negative predictive value (NPV) were used. The MetS prevalence was 27.4%, which means that 74 (38.0 ± 7.3 years) military men had at least three risk factors of MetS present. The cutoff point of VAT with the highest balance between sensitivity (77.0%) and specificity (69.9%) was 1025.0 cm^3^ (1086.0 g). An area on the ROC curve was 0.801 (*p* < 0.000), which was very good precision. VAT ≥ 1025.0 cm^3^ (1086.0 g) is associated with the risk factors of MetS and is, therefore, a predictor of the disease with good indicators of sensitivity and specificity and a robust indicator of MetS.

## 1. Introduction

Obesity is characterized by excessive accumulation of adipose tissue with damage to health and it is considered a worldwide epidemic conditioned mainly by the food and physical activity profile [1]. It is also the main risk factor for Chronic Non-communicable Diseases (NCDs) such as diabetes, stroke, hypertension, cardiovascular diseases and Metabolic Syndrome (MetS) [1]. The global prevalence of MetS in the population varies according to the diagnostic criterion used and geographic and sociodemographic factors [2,3]. The latest consensus, Joint Interim Statement (JIS), pointed out that about 20% to 25% of the world’s adult civilian population has three of the five risk factors for the disease [4]. A recent study, using JIS criteria and involving 2719 (27.7 ± 8.59 years) male military selected to participate in the United Nations peacekeeping mission in HAITI, showed that 12.2% of the soldiers had risk factors for MetS [5].

Scientific evidence shows that MetS is associated with obesity, mainly with the accumulation of visceral adipose tissue (VAT), and consequently with cardiovascular risk [3,6]. Even if the subcutaneous adipose tissue (SAT) represents about 80% of the total body fat, it is VAT that has greater metabolic activity secreting a large number of adipokines, such as Leptin, which are related to inflammatory processes [7]. Furthermore, VAT is also considered as an independent risk factor for pathophysiological changes in MetS [3]. Thus, quantifying the VAT may be key to assess the risk of developing NCDs, including metabolic complications [3,7,8].

Some anthropometric measurements can estimate the amount of VAT, including waist circumference (WC) and associate it with risk factors, but these present limitations [9]. However, imaging techniques such as computed tomography (CT) and magnetic imaging resonance (MIR) can be accurately measured VAT and associated with some risk factors [10]. However, they expose the patient to high radiation [10]. The view of this dual X-ray absorptiometry (DXA) is a reference method for assessing body composition with the advantages of high precision, low radiation and short scanning time [11,12].

Despite the fact that some studies have investigated the association between excess VAT and the association with cardio-metabolic risk, few have proposed VAT cutoff points using DXA [11,12,13,14,15,16,17,18,19]. Only two studies, using GE’s DXA, were found to associate VAT with cardio-metabolic risk factors [18,19]. However, no study involving VAT cut-off points associated with MetS in militaries has been found. A cutoff point can facilitate the diagnosis of people with the risk factors for metabolic syndrome, preventing them from being subjected to biochemical tests, blood pressure assessment and anthropometric measurements.

In this context, the objective of the present study was to determine the VAT cutoff points assessed by “DXA” associated with MetS in military men.

## 2. Materials and Methods

### 2.1. Study Design and Population

This is a cross-sectional study carried out in a convenience sample of 270 male military men, aged 22 to 60 (37.5 ± 6.9) years. They were selected from the group of volunteer military personnel who participated in the MetS Monitoring Program conducted by the Brazilian Army Physical Fitness Research Institute (IPCFEx) between 2017 and 2019. In this program, all military men received a complete health assessment by specialized professionals from IPCFEx. Active-duty male military personnel who were students at the Army Command and General Staff School (ECEME), the Officer Improvement School (EsAO) and the Sergeants School of Logistics (EsLog) were included. In addition, militaries selected to integrate individual peacekeeping missions of the United Nations and militaries from BA Organizations in the city of Rio de Janeiro were included. Military retired personnel and those who underwent any type of recent abdominal surgery were excluded. Female military personnel were also excluded due to the small number evaluated (only 27 women had participated until the end). This is illustrated in Figure 1.

All the participants signed the informed consent form when they became aware of the procedures and risks to which they would be submitted. This research was registered in the National Research Ethics System, submitted and approved by the Ethics Council of the Naval Marcílio Dias Hospital, n° 1.551.242, CAAE n° 47835615.5.0000.5256 on 11 July 2019.

### 2.2. Data Collect

This took place in the IPCFEx laboratories in four stages during the same day in the morning, with the participant fasting for 12 h overnight. Blood samples were included for biochemical analysis in the IPCFEX Clinical Analysis Laboratory. Blood pressure assessment, anthropometric and body composition evaluation by DXA were realized in the IPCFEX Body Composition Laboratory.

### 2.3. Blood Collection and Biochemical Analyzes

Initially, blood samples (8 mL) were collected, in vacutainer tubes without anticoagulant, by a trained professional from 7 to 9 a.m. The blood samples were analyzed immediately after collection. Serum aliquots were separated after centrifugation (3000 rpm for 15 min) in the device model Centurion vector inverter system with 32 tubes (Centurion Scientific, Stoughton, Chichester, UK). The concentrations of triglycerides (TG), glucose (GL) and high-density lipoprotein (HDL-c) were determined using a biochemistry analyzer, model bt 3000 plus (Wiener Lab, Rosário, BA, Argentina). The serum levels were analyzed using the endpoint enzymatic colorimetric methodology, with GOD-PAP (glucose oxidase + peroxidase) by spectrophotometry, in the case of GL, GPO/PAP (glycerol phosphate oxidase + peroxidase), in the case of TG and colorimetric without precipitation CHO/PAD/TOOS (N-ethyl-N-3-disodium toluidine), in the case of HDL-c. The collections followed all the recommendations of the Brazilian Society of Clinical Pathology/Laboratory Medicine and according to Resolution n° 306/2004 from National Health Surveillance Agency for the management and disposal of samples [20,21].

### 2.4. Blood Pressure Assessment

Afterwards, Systolic Blood Pressure (SBP) and Diastolic Blood Pressure (DBP) were measured according to the Brazilian Guideline of Arterial Hypertension for the Use of Ambulatory Blood Pressure Monitoring of the Brazilian Society of Cardiology [22].

### 2.5. Anthropometric Evaluation

Then, military men underwent an anthropometric assessment, conducted by two physical education professionals from the IPCFEx, following the protocol standardized by Fernandes Filho [23]. The technical errors of measurement inter and intra-evaluators were considered acceptable, being 1.35% and 0.97%, respectively [24]. Total body mass (BM) was determined with those evaluated in the orthostatic position (OP), barefoot, using only a bathing suit a digital scale, model P150M (Lider, Araçatura, São Paulo, Brazil), with a maximum load of 200 kg. Height was measured with a metal stadiometer (Sanny, São Bernarndo do Campo, São Paulo, Brazil), with an accuracy of 2 mm. The subjects were standing, in OP, keeping contact with the device by the heel. To measure the WC, an anthropometric measuring tape, model SN 4010 (Sanny, São Bernarndo do Campo, São Paulo, Brazil), was used, with those evaluated in an OP and the abdomen relaxed. WC was measured at the point with the lowest circumference line of the last rib with the tape placed in a horizontal plane and with the individual in respiratory apnea, to minimize possible instability in the spine.

### 2.6. DXA Body Composition

The body composition assessment was performed using DXA, iLunar model, from GE Healthcare (GE Healhcare, Madison, WI, USA), with the enCore 2015 software (version 14.10.022). Before each acquisition DXA scanner was calibrated according to the manufacturer’s instructions [12,14,16]. Furthermore, the calibration column phantom was applied weekly. From the scan of the whole body, the data of total fat mass (FM), total lean mass (LM), percentage of total body fat (%BF-DXA) and the fat mass index (FMI), FM (kg)/Height^2^ (m^2^), were obtained. VAT was measured and a region-of-interest is automatically defined whose caudal limit is placed at the top of the iliac crest and its height is set to 20% of the distance from the top of the iliac crest to the base of the skull to define its cephalad limit [12,14,16]. VAT mass (g) was automatically transposed into volume (cm^3^) using a constant correction factor (0.94 g/mL). The measurements in the calibration block (daily) presented acceptable coefficients of variation 1.0% [12,14,16].

### 2.7. MetS Diagnosis

The diagnosis of MetS followed the parameters of the JIS with a combination of three or more cutoff points: central obesity (WC men ≥ 90 cm), HDL-c (men ≤ 40 mg/dL and/or use of medication for control), TG (men ≥ 150 mg/dL and/or use of medication for control), GL (men ≥ 100 mg/dL and or use of medication for control) and high levels of SBP ≥ 130 mmHg and/or DBP ≥ 85 mmHg and/or use of medication for control [4].

### 2.8. Statistical Analysis

Descriptive statistics were performed to characterize the sample universe surveyed using the measures of location (mean) and dispersion (SD). The Kolmogorov–Smirnov test was used to test the normality of the data distribution. In this sense, a parametric analysis was used. The Student’s *t*-test, from independent samples, compared the anthropometric, hemodynamic, biochemical and body composition variables of the DXA among the military personnel without MetS risk factors (WMetSRF) and with the referred risk factors (MetSRF). The variation coefficient percentage (%CV) was calculated using the formula: %CV = (standard deviation (SD)/mean) × 100. To measure the reliability of measurements it was calculated the intra-class correlation coefficient (ICC) and Cronbach’s alpha coefficients (α). The effect size was calculated by Cohen’s d [25]. The Pearson correlation test was used to verify the association with the VAT obtained by DXA (VAT-DXA). The Receiver Operator Characteristic (ROC) curve was constructed to establish the VAT cutoff points. The Youden Index was used to determine the performance of the test and to determine more precisely which cutoff point of VAT with more balance [26]. The positive predictive values (PPV) and negative predictive values (NPV) were defined, respectively, as the probability that the military present the risk factors for MetS based the VAT equal to or above the cutoff point and does not present the risk factors by the VAT below the cutoff point [27]. The level of significance adopted was 95% (*p* < 0.05) and statistical analyses were performed by SPSS version 17 (Armonk, Nova York, NY, USA).

## 3. Results

After applying the criteria for the diagnosis of MetS, 74 militaries were diagnosed with the disease, a prevalence of 27.4%. Table 1 presents the general descriptive data of the 270 male military personnel evaluated and the comparison between the MetSRF and WMetSRF groups. Military personnel diagnosed with MetS had significantly higher anthropometric, hemodynamic, body composition and biochemical measurements than the group without the diagnosis (*p* < 0.05). Height alone was not significantly different between groups. Furthermore, the size of the effect in most of parameters was “large” (d = 0.80–1.29) [25]. Only four parameters showed “medium” effect size (d = 0.50–0.79) [25].

Table 2 shows the associations among anthropometric, hemodynamic, biochemical, body composition and VAT-DXA (cm^3^) variables by groups. In the group without risk factors for the disease, strong and significant associations were observed between VAT and BM, FM, BMI, FMI, %BF-DXA and WC. This group also presents significant associations between VAT and age, LM, TG and glucose. VAT was negatively associated with HDL-c concentrations. In the MetSRF group VAT had strong association with FMI, FM, %BF-DXA and WC and moderate association with BM and BMI. In addition, it presented a weak association with age and glucose. In both groups, no associations were found between VAT, height, SBP and DBP.

VAT cutoff points with degrees of sensitivity and specificity and predictive values are shown in Table 3. A VAT cutoff point that showed best balance in the specificity and sensitivity test was 1025 cm^3^, which corresponded to 1086 g, with a sensitivity of 77.0%, specificity of 69.9%, PPV of 49.1% and NPV of 89.0%.

This statement can be seen in the analysis of the ROC curve (Figure 2), which revealed, with the help of the Youden Index, PPV and NPV, that the VAT cutoff point with the greatest balance between sensitivity had an area on the ROC curve was 0.801 (*p* < 0.001) [26,27].

## 4. Discussion

The present study determined the VAT cutoff points assessed by DXA associated with MetS in BA military men. At the end, it was observed that the VAT ≥ 1025 cm^3^, which is equivalent to 1086 g, showed a better ability to predict the disease in these individuals.

The prevalence of MetS has grown exponentially in the world and in Brazil, even in military personnel that have good physical conditioning [5,28,29,30]. Recently, Rostami and collaborators (2019) postulated, after a systematic review with meta-analysis, that the average prevalence of MetS in military personnel from the Armed Forces of the four continents is 8.0% [30]. They also suggested that the prevalence in this group is lower when compared to the civilian population, because they are encouraged to have a healthier lifestyle and to perform physical training with regular frequency [30].

In Brazil, a survey involving military and MetS analyzed 1383 military personnel from the Brazilian Navy, in active service, and concluded that the prevalence of MetS was 17.6%, being the HDL-c the most prevalent risk factor in the population, which was present in 43% of the individuals, followed by high blood pressure (26.3%), hypertriglyceridemia (19.3%) and fasting glucose (6.6%) [29]. Likewise, Fortes et al. postulated that the prevalence of MetS in 2719 (27.7 years) Brazilian military personnel selected to participate in the UN peacekeeping mission in HAITI, from 2014 to 2017, was 12.2%, using the JIS criteria [4]. They also stated that both WC and BMI can be considered as good predictors of changes in the physiological markers of MS, especially in the military [4]. This prevalence was lower in relation to that found in the article under discussion (27.4%) whose average age was 38.0 years, therefore higher than in the surveys mentioned above. Perhaps here is the answer to this difference, as the prevalence of MetS increases with age, regardless of the diagnostic criteria used [2,3,4,30].

It is well described in the literature that biochemical, anthropometric and blood pressure markers are altered in the presence of MetS [5,29,30,31]. Rosa and collaborators found significantly lower mean values of WC, HDL-c, SBP and DBP in Brazilian military personnel without MetS risk factors, indicating a lower risk of DCNs [28].

Gámez et al. conducted a prospective cross-sectional survey with 51 Cuban military personnel MetSRF and 43 WMetSRF and found that %BF, HLD-c, glucose, TG, SBP, DBP and WC variables showed greater differences between group means. Similar to the findings described above, the study under discussion suggests that military personnel diagnosed with MetS risk factors have significantly worse health indicators when compared to individuals without risk [31]. The data obtained in this study confirm those described in a recent systematic review, suggesting that people diagnosed with MetS have a worsening in quality of life when compared to those who presented the disease [32].

In recent years, DXA has been a reliable alternative to CT and MRI to estimate total and regional fat [12,17,18]. In the present research, the VAT found in the individuals with MetS was significantly higher compared to the group WMetSRF; in fact, it was almost double. Current findings describe the occurrence of an association between VAT, measured by DXA, with cardiac and metabolic risk factors including high levels of glucose, hypertension, increased WC, BMI and high %BF, but none of the studies evaluated this outcome in the military population [6,7,12,13,14,15,16,17,18,19].

On the other hand, Sasai et al. prove that the VAT of adults, with BMI ranges from 19 to 54 kg/m^2^, was inversely associated with HDL-c and positively with TG, WC, %BF and BMI, corroborating the finding in the present study [8]. Nicklas and collaborators suggested that the VAT-DXA of American women, aged 45 to 73 years, is positively correlated with WC and that, in turn, is related to all risk factors for MetS, except total and low cholesterol density lipoproteins (LDL) [17]. Another research study showed that the VAT-DXA (cm^3^) of 229 women, diagnosed with obesity, was associated with the variables of WC, SBP, DBP, HDL-c, TG and glucose [18]. In the military, although research involving VAT and risk factors for MetS was not found, Rosa and collaborators carried out a study with 262 male soldiers, aged between 19 and 49 years, selected for the peace mission of the UN in Haiti, using GE’s DXA, iLunar model, and observed an inverse association between HDL-c and EC, showing indirect instruments that are independent predictors of NCDs [33].

Although research points to the occurrence of associations between VAT and BP, in the present article, there were not associations between VAT and BP in both groups [34]. Differently from what literature describes, in the group with MetS it was not possible to observe a negative association between VAT and HDL-c concentration. This association was only observed in the group without disease. It is possible that the change in HDL-c does not accompany the change in VAT, justifying the absence of association in this group.

Considering the scientific literature studied, VAT cutoff points were not found; these were measured by DXA, in cm^3^, and associated with MetS in the active military population. The first cutoff point associated with metabolic and coronary risk was established by Nicklas and collaborators had used GE’s DXA, model DPX-L Lunar, which presents VAT in area (cm^2^) [18]. The results of this study showed that women, with a mean of 59.0 ± 6.0 years, had a higher risk of having metabolic and coronary problems if they had a VAT ≥ 163 cm^2^.

Ten years later, Kelly and collaborators established VAT cutoff points, using the DXA, Hologic Discovery, also reading in area (cm^2^) and suggested that women with VAT ≥ 100 cm^2^ would have an increased metabolic risk for coronary heart disease, while those with VAT ≥ 160 cm^2^ had an even higher metabolic risk [35].

Only two studies determined VAT cutoff points, in volume (cm^3^) associated with MetS risk factors using the GE’s DXA [18,19].

In the first, the researchers obtained cutoff points for women of European and African American descent (BMI 30.0 kg/m^2^), using the ROC curve (area under curve, AUC, 0.749) with a balance between sensitivity and specificity (50%) [18]. This research postulated cardio-metabolic risk associations for European women with a VAT ≥ 1713 cm^3^ and for African American women a VAT ≥ 1320 cm^3^ [18]. In another finding, involving 421 young Europeans of both sexes, aged 20 to 30 years, it was observed that, in men, VAT ≥ 761 cm^3^ was associated with risk factors for cardio-metabolic and VAT ≥ 239 cm^3^ for women [19]. The area under the ROC curve was 0.914 for men and 0.839 for women [19].

As you can see, the different cutoff points attributed so far vary from one another and are different from the one found in this article. Part of this difference is attributed to the way of quantifying the VAT, being in area or volume and to the different characteristics of the studied populations [17,18,19,31]. When interpreting the area under the ROC curve found in the current article, whose value was 0.801 (*p* < 0.001), it is observed that it has very good diagnostic accuracy [26,27]. The Youden Index, measuring the performance of the test, calculated by deducting 1 from the sum of sensitivity and specificity of the test, helped in the discovery of the cutoff point with more balanced sensitivity (77.0%) and specificity (69.9%) (Table 2) [15,26,27].

In this study the VAT value ≥ 1025 cm^3^ (1086 g) showed a better balance relationship with PPV and NPV. This means that if the military has visceral adiposity ≥1025 cm^3^ he has a 49.1% chance of having MetS, and if he has a VAT < 1025 cm^3^ he has 89.0% chance of not having MetS.

Unlike sensitivity and specificity, predictive values (PV) are dependent on the prevalence of the disease in the population examined. Therefore, the PV should not be transferred to another population with a different prevalence of the disease. Prevalence affects PPV and NPV differently. The PPV is increasing, while the NPV decreases with the increase in the prevalence of the disease in the population. These values of PPV and NPV are in a medium zone where they have the highest efficiency and the best cost–benefit ratio [27]. A possible explanation for this is in the sample size, as the 57 individuals who presented risk factors for MetS also had a VAT ≥ 1025 cm^3^. However, no studies were found in the literature that involved the positive and negative prediction of MetS and the amount of visceral adiposity.

It is relevant to understand that the simple comparison of VAT values over time does not give information on the risk category to which the subject is exposed. With a cut-off point as a reference, the subjects can have a way to estimate their situation regarding the risk of being in the condition of MetS with only one exam (DXA). This cut-off point will also serve as a strategy to complete clinical information in the diagnosis of diseases, especially in subjects who are facing some metabolic disorders related to excess fat.

It is important to recognize some limitations in the present study: VAT-DXA cutoff points associated with risk factors for MetS were investigated exclusively in men; the generalization of the results was restricted to the military men aged between 22 and 60. Even so, this article presents a strong point because it is the first study that determined VAT cutoff points (cm^3^), assessed by DXA, GE iLunar, associated with MetS in military personnel from BA. It is important to highlight that no other similar study was found in the scientific literature, even in other populations.

## 5. Conclusions

The results of the present study suggest that VAT ≥ 1025.0 cm^3^ (1086.0 g), determined by DXA, GE iLunar, is associated with risk factors for the disease and can, therefore, predict it with good indicators of sensitivity and specificity.

These findings may be can be extrapolated to the Brazilian male population, as the BA is an institution formed by a heterogeneous sample of Brazilian society. Brazilians from different regions of the country join the Brazilian army in different ways. Whether by mandatory military conscription, or by public contests held annually. For this reason, such findings can be used, in daily clinical practice, by all male Brazilians within the studied age range.

They can also serve as a support for future investigations and treatments of MetS, improving the health and quality of life of the military, thus increasing its operability.

## Figures and Tables

**Figure 1 healthcare-09-00886-f001:**
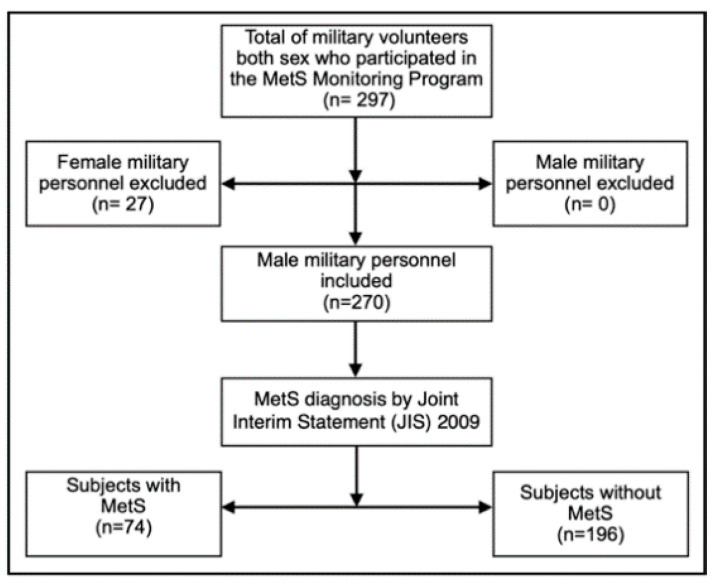
Schematic description of the study, with subject who participated of the Metabolic Syndrome (MetS) Monitoring Program conducted by the Brazilian Army Physical Fitness Research Institute (IPCFEx).

**Figure 2 healthcare-09-00886-f002:**
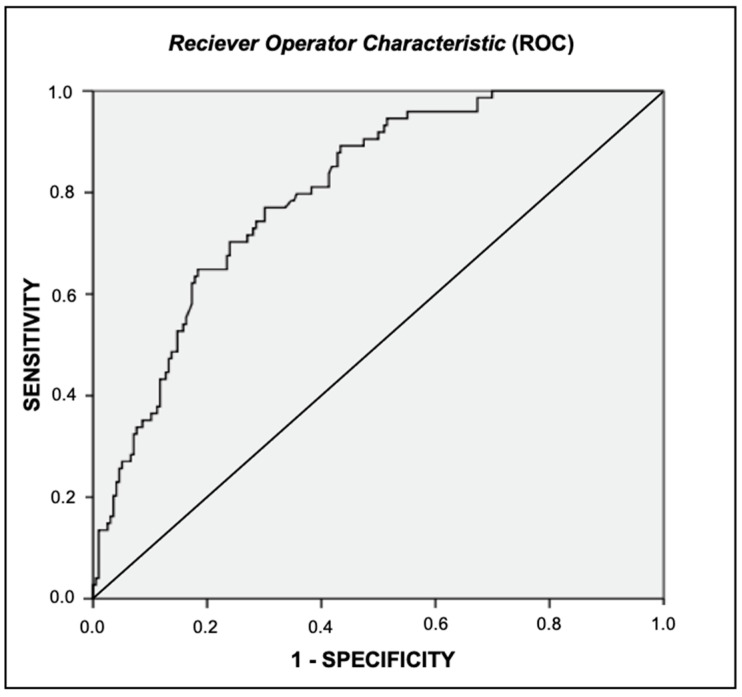
ROC curve for cut-off point of visceral adipose tissue associated with Metabolic Syndrome.

**Table 1 healthcare-09-00886-t001:** Descriptive data of the military evaluated and comparison between groups (*n* = 270).

Parameters (*n*)	WMetSFR (196)	MetSFR (74)	*p* Value *	Effect Size (d)	95%CI	ICC	CV	*α*
	Mean ± SD	Mean ± SD						
Age	36.0 ± 6.3	38.0 ± 7.3	0.026	0.30	0.04–0.57	0.421	1.71	0.448
Height (cm)	177.5 ± 6.0	177.5 ± 6.2	0.829	0.02	0.23–0.29	−0.061	1.61	−0.060
BM (kg)	80.9 ± 12.3	94.5 ± 12.3	0.000	0.98	0.70–1.26	0.084	2.18	0.108
BMI	25.7 ± 3.6	29.8 ± 6.2	0.000	1.03	0.76–1.32	0.150	6.61	0.197
LM (kg)	58.8 ± 5.9	62.5 ± 6.5	0.000	0.60	0.33–0.88	−0.038	0.36	−0.041
FM (kg)	20.8 ± 8.4	28.9 ± 8.7	0.000	0.95	0.67–1.23	0.024	10.28	0.030
FMI (kg/m^2^)	6.6 ± 2.6	9.2 ± 2.7	0.000	0.97	0.69–1.25	0.044	13.30	0.057
%BF-DXA	24.5 ± 6.7	30.1 ± 5.7	0.000	0.87	0.59–1.15	0.029	8.41	0.037
VAT (cm^3^)	806.4 ± 597.6	1550.6 ± 716.8	0.000	1.18	0.89–1.46	−0.189	27.87	−0.296
VAT (g)	761.2 ± 563.9	1462.8 ± 676.2	0.000	1.18	0.89–1.46	−0.189	27.85	−0.296
WC (cm)	87.1 ± 8.6	96.3 ± 7.6	0.000	1.14	0.85–1.42	−0.087	1.98	0.121
Triglycerides (mmol/L)	83.4 ± 38.0	117.8 ± 55.7	0.000	0.78	0.51–1.06	0.011	1.71	0.014
Glucose (mmol/L)	94.5 ± 8.6	100.7 ± 10.0	0.000	0.58	0.41–0.96	−0.361	0.82	−0.367
HDL-c (mmol/L)	54.4 ± 16.0	46.9 ± 10.8	0.000	0.52	0.24–0.79	0.087	8.08	0.101
SBP (mmHg)	117.1 ± 9.7	130.0 ± 12.8	0.000	1.21	0.92–1.49	−0.319	1.56	−0.628
DBP (mmHg)	80.0 ± 6.2	90.0 ± 11.3	0.000	1.28	0.98–1.56	−0.007	4.80	−0.010

WMetSRF: military group without metabolic syndrome, MetSRF: military group with metabolic syndrome, SD: standard deviation, %CV: Coefficient of variation, ICC: intra-class correlation coefficient, α: Cronbach’s alpha coefficients, BMI: body mass index, WC: waist circumference, BM: total body mass, FM: total fat mass, LM: total lean mass, FMI: fat mass index, %BF-DXA: percentage body fat, VAT: visceral adipose tissue, HDL-c: high-density lipoprotein, SBP: systolic blood pressure, DBP: diastolic blood pressure. DXA: dual X-ray absorciometry. * *p* value obtained by Student’s *t*-test. (d) Cohen’s d Effect Size.

**Table 2 healthcare-09-00886-t002:** Correlation between VAT-DXA and independent variables of militaries without MetS and with MetS.

Variables	VAT-DXA			
	WMetSRF		MetSRF	
Independent Variables	r *	*p* *	r *	*p* *
Age	0.280	0.000	0.363	0.000
Height (cm)	0.071	0.323	0.124	0.291
BM (kg)	0.775	0.000	0.548	0.000
BMI	0.800	0.000	0.644	0.000
FMI (kg/m^2^)	0.872	0.000	0.747	0.000
LM (kg)	0.331	0.000	0.056	0.637
FM (kg)	0.880	0.000	0.722	0.000
%BF-DXA	0.825	0.000	0.722	0.000
WC (cm)	0.885	0.000	0.836	0.000
Triglycerides (mmol/L)	0.260	0.000	0.164	0.163
Glucose (mmol/L)	0.216	0.002	0.343	0.003
HDL-c (mmol/L)	−0.175	0.014	−0.138	0.241
SBP (mmHg)	0.033	0.643	0.019	0.874
DBP (mmHg)	0.119	0.098	0.127	0.282

WMetSRF: military group without metabolic syndrome, MetSRF: military group with metabolic syndrome, BM: total body mass, FM: total fat mass, LM: total lean mass, FMI: fat mass index, %BF-DXA: percentage body fat DXA, VAT: visceral adipose tissue, WC: waist circumference, HDL-c: high-density lipoprotein, SBP: systolic blood pressure, DBP: diastolic blood pressure, * *p* value obtained by Person correlation.

**Table 3 healthcare-09-00886-t003:** VAT cutoff points with other relationships of sensitivity, specificity and predictive values.

VAT Cutoff (cm^3^)	Sensitivity (%)	Specificity (%)	Youden Index **	PPV (%)	NPV (%)
48.0	100.0	0.0	0.00	27.4	0.0
568.5	95.9	44.9	0.41	39.7	3.3
811.5	85.1	57.7	0.43	43.3	91.1
**1025.0 ***	**77.0**	**69.9**	**0.47**	**49.1**	**89.0**
1417.0	56.8	83.2	0.40	56.0	83.6
2062.5	23.0	95.9	0.35	69.2	77.0
4201.0	0.0	100.0	0.00	27.2	100.0

VAT: visceral adipose tissue, * more balanced VAT cutoff found, ** index evaluation of the overall discriminative power of a diagnostic procedure, PPV: positive predictive value, NPV: negative predictive value.

## Data Availability

All data generated in this study are contained within this publication.

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
