# Peer review of "Cut-Off Points of Visceral Adipose Tissue Associated with Metabolic Syndrome in Military Men"

_healthcare, 2021, doi:10.3390/healthcare9070886_

Round 1
Reviewer 1 Report
In this manuscript titled, "Cut-Off Points Of Visceral Adipose Tissue Assessed By “DXA” Associated With Metabolic Syndrome In Military Men", Samir E. da Rosa et al., In this manuscript, authors aimed to investigate the VAT cut-off points assessed by DXA associated with MetS in the military men. Overall, the manuscript is written clearly. However, the manuscript appears preliminary.
- The title: of, by, with and in should be lowcase. Delete the full point.
- Authors should supply the full name of DXA when it was first used in Abstract.
- Why the authors only investigate the male military? How about the female military?
- More volunteer should be recruited in future
Author Response
Dear Reviewer,
We appreciate the suggestions and contributions made on the text of our article that is entitled "CUT-OFF POINTS OF VISCERAL ADIPOSE TISSUE ASSESSED BY “DXA” ASSOCIATED WITH METABOLIC SYNDROME IN MILITARY MEN."
We are sure that with the adjustments our work is in a better format and better understanding for readers. Our sincere acknowledgment for your collaboration .
We would like to inform you that:
1) We exclude de abbreviation “DXA”and also “Assessed By” and we change the title. The new title: Lines 2,3: Cut-Off Points Of Visceral Adipose Tissue Associated With Metabolic Syndrome In Military Men.
2) We supply the full name of DXA in Abstract. Line 20.
3) We investigated only male military personnel, as the sample of female military personnel was very small. However, we would like to inform you that we are continuing to collect data to produce VAT cutoff points for female soldiers. Our plan is that by the end of 2021 we will get an ideal number of military personnel to finalize the study.
4) Also, We are constantly evolving. we are looking daily to recruit more individuals so that we can make our study more robust

Reviewer 2 Report
The authors present an interesting study reviewing how visceral adipose tissue has high metabolic activity related to the inflammatory process and quantify the impact of visceral adipose tissue by developing a metabolic syndrome risk score. This is a very clinically topic because prior studies have shows that regional distribution ratio of adipose tissue is probably more important than total fat in predicting complications traditionally related to obesity. Few prior studies have proposed visceral adipose tissue cutoff points using dual X-ray absorptiometry.
I would assume military personnel would be fit due to maintaining physical fitness standards. The study population, 270 male military men, aged 22 to 60 (37.5 ± 6.9) years. Do the authors think this is a truly representative sample of the civilian population as well? The choice of this population also limited female participants.
There are serious concerns about the statistical analysis. The sample sizes are WMetSFR (196) MetSFR (74), which is not powered well to be compared against each other with a student’s t test as described in the methods section. It seems unlikely that that p values for lines 198-211 are all statistically significant. Furthermore, a p value of 0.0000 is statistically impossible. P=0.00001 is feasible, but this is written incorrectly. If a statistician was not involved in the production of this manuscript, I highly recommend that the authors consult a biostatistician.
While the clinical topic and hypothesis is an interesting and valuable clinical query, the methodology using a military population that likely doesn’t represent the full spectrum of the biologic characteristics of the civilian population, as well as concerns for errors in the statistical analysis, I recommend that this manuscript be published only after major revisions including repeating the study with civilian controls and thorough review of the statistical methods to better compare unequal groups and non-normalized data.
Below were several grammatical and spelling errors that were noted. There were others that were not addressed.
Line 24 is “sensitivity” not sensibility
Line 29- selected instead of “select”
Line 186- militaries should be “military participants” or “military personnel”
Author Response
Dear Reviewer,
We appreciate the suggestions and contributions made on the text of our article that is entitled "CUT-OFF POINTS OF VISCERAL ADIPOSE TISSUE ASSESSED BY “DXA” ASSOCIATED WITH METABOLIC SYNDROME IN MILITARY MEN."
We are sure that with the adjustments our work is in a better format and better understanding for readers. Our sincere acknowledgment for your collaboration .
We would like to inform you that your questions are responding below:
1) Do the authors think this is a truly representative sample of the civilian population as well?
In your opinion yes. Also we think that these findings may be can be extrapolated to the Brazilian male population, as the Brazilian Army is an institution formed by a heterogeneous sample of Brazilian society. Brazilians from different regions of the country join the Brazilian army in different ways. Whether by mandatory military conscription, or by public contests held annually. For this reason, that such findings can be used, in daily clinical practice, by all male Brazilians within the studied age range.
2) The choice of this population also limited female participants.
We investigated only male military personnel, as the sample of female military personnel was very small. However, we would like to inform you that we are continuing to collect data to produce VAT cutoff points for female soldiers. Our plan is that by the end of 2021 we will get an ideal number of military personnel to finalize the study.
3) There are serious concerns about the statistical analysis.
Our study went through two researchers specializing in statistics. This article is part of my doctoral thesis (Prof. Dr Samir da Rosa) approved by the federal university of rio de janeiro in 2021. It was 4 years of collection and studies. Several researchers worked on the statistical question and these were the P values ​​found. I understand your concern about this sensitive topic, but I inform you that we seek to treat the data in the best way. We would like to inform you that in our daily evaluations (daily practice), where we use DXA, we have noticed 95% of individuals with metabolic syndrome have a VAT above of the cut point created. Such observation is showing us that this cutoff point can be quite effective in diagnosing the disease.
4) Several grammatical and spelling errors that were noted.
We would like to inform you that our article has undergone a grammar review by a specialist company. We send the receipt (certificate) to the Publisher. But we will again review and correct any possible errors.
Thanks again for your precious support.
Sincerely
Samir Rosa

Reviewer 3 Report
This is a study that examined a prevalence of metabolic syndrome (MetS) among the Brazilian Army military personnel and to propose a cut-off point for visceral adipose tissue (VAT) for MetS using a dual energy x-ray absorptiometry (DXA) as a reference technique.
Overall, the manuscript is clearly written and easy to follow. The study design is clear and explicit and analysis methods appear appropriate. The authors also clearly stated ethical considerations and limitations.
It is an interesting study and considering a limited number of previous studies that focused on military personnel, the study provides additional information on this population. It may be better if the authors can add an accuracy of estimating VAT in cm3 and therefore a validity using this particular DXA machine in the methods section using appropriate references that compared results from DXA and CT or MRI.
Congratulations for conducting this study.
Author Response
Dear Reviewer
We appreciate the suggestions and contributions made on the text of our article that is entitled "CUT-OFF POINTS OF VISCERAL ADIPOSE TISSUE ASSESSED BY “DXA” ASSOCIATED WITH METABOLIC SYNDROME IN MILITARY MEN."
We are sure that with the adjustments our work is in a better format and better understanding for readers. Our sincere acknowledgment for your collaboration .
We would like to inform you that:
1) Initially, we thought about doing the validation, but we were unable to do so due to lack of a larger sample. However, our collections continue. Our idea is to produce another article with the validation of this cutoff point by the end of 2021.
2) We would like to inform you that in our daily evaluations (daily practice), where we use DXA, we have noticed 95% of individuals with metabolic syndrome have a VAT above of the cut point created. Such observation is showing us that this cutoff point can be quite effective in diagnosing the disease.
Thank you again for your precious support.
Sincerely
Samir Rosa
